

# 3D-printed Ag-AgCl Electrodes for Laboratory Measurements of Self-Potential

Thomas Simon Leacroft Rowan[1*], Vilelmini A. Karantoni[2] Adrian P. Butler[1], M. D. Jackson[2]

[1]Department of Civil and Environmental Engineering, Imperial College London, UK.

[2]Department of Earth Science and Engineering, Imperial College London UK.

*Correspondence to*: Tom Rowan (t.rowan@imperial.ac.uk)

**Abstract** *This paper details the design, development and evaluation of a 3D printed, rechargeable, Ag-AgCl electrode to measure Self-Potential (SP) in laboratory experiments. The challenge was to make a small, cheap, robust and stable electrode that could be used in a wide range of applications. The new electrodes are shown to offer comparable performance with*

*custom-machined laboratory standards, and the inclusion of 3D printing (both Fused Filament Fabrication (FFF) and stereolithography (SLA)) makes them more versatile and less expensive than laboratory standards. The devices have been used in both low-pressure experiments using beadpacks, and high-pressure experiments using natural rock samples. Designs are included for both male and female connections to laboratory equipment.*

## 1 Introduction

Measurements of Self-Potential (SP) are used by environmental and engineering site surveyors (Nyquist and Corry, 2002; Weigand et al., 2020; Eppelbaum, 2021), hydrologists (Graham et al., 2018; Macallister et al., 2019; Maineult et al., 2008; Revil et al., 2003; Rizzo et al., 2004; Sailhac and Gibert, 2003), and for monitoring volcanic and seismic activity (Aubert and Atangana, 1996; Finizola et al., 2004; Ishido, 1989). Additionally, SP is utilized for leakage detection from dams and embankments (Bogoslovsky and Ogilvy, 1973; Bolèkve et al., 2009; Ogilvet et al., 1969), locating and monitoring contaminant

plumes (Linde and Revil, 2007; Minsley et al., 2007; Naudet et al., 2004; Naudet et al., 2003), detecting subsurface voids, disturbances and sinkholes (Jardani et al., 2006; Jardani et al., 2007; Eppelbaum, 2020), and monitoring pumping and sparging tests (Jackson, et al., 2012; Maineult et al., 2008; Rizzo et al., 2004).

Nonpolarizing electrodes, consisting of a metal (Cu, Ag, or Pb) immersed in a metal salt solution or coated with a metal salt and immersed in a conductive solution (Ag–AgCl or Pb–PbCl2 in NaCl or KCl; Jackson, 2015), are commonly used in SP

data acquisition. To reduce leakage of the electrolyte, a low-permeability membrane (ceramic) may be employed, or a gelling agent or solid medium (kaolinite or plaster) may be used (Jackson, 2015). In some cases, the reference electrolyte may be natural (e.g., seawater; Jackson, 2015).

Nonpolarizing electrodes are characterized by limited electrode polarization and drift, due to the nearly equal magnitude of polarization at the metal–electrolyte interface in each electrode (Jackson, 2015). This causes the polarization across an

electrode pair to be cancelled out and only vary slowly in response to shifts in the reference electrolyte composition and concentration. This type of electrode, also known as 'liquid-junction' or 'reference' electrodes, is not truly nonpolarizing, but the term is often used to describe their behaviour (Jackson, 2015). The 'effective' electrode polarization of these electrodes is the diffusion (liquid-junction) potential across the contact between the reference electrolyte solution and the adjacent medium.



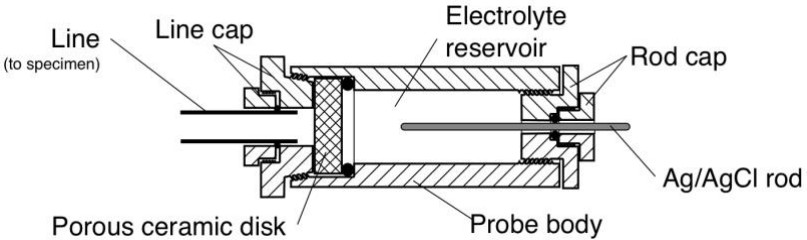

**Figure 1: Parts of a typical rechargeable, non-polarising Ag-AgCl electrode for laboratory SP measurements. The 'rod cap' holds a silver rod in place in the electrolyte reservoir and allows electrical connection to the rod. The 'line cap' connects the electrode to the experiment via a fluid-saturated flow line. A porous ceramic disk allows electrical charge exchange between the electrolyte and the experimental fluid without mixing of the fluids. The rod cap can be removed to allow the silver to be replaced, and the line cap can be removed to allow the ceramic disk and/or the electrolyte to be replaced. Hence the electrode can be periodically 'recharged' as described in the text. Modified from Vinogradov et al., 2010)).**

Comparative studies of nonpolarizing electrodes for geophysical applications demonstrate that the performance of each electrode type depends on the measurement conditions and that no one type outperforms all others (Jackson, 2015; Perrier et al., 1997; Petiau and Dupis, 1980). Ag–AgCl, Cu–CuSO$_4$ and Pb–PbCl$_2$ electrodes are the most widely used and feature a metal electrode and electrolyte contained within a ceramic or plastic casing, with a ceramic or wood membrane providing the electric connection (Corry et al., 1983; Corwin, 1980; Jackson, 2015; Jackson et al., 2012; Maineult et al., 2008; Perrier, Frédéric and Pant, 2005; Petiau, 2000; Vinogradov, J. et al., 2010). Commercial suppliers have begun to manufacture electrodes suitable for SP measurements in the field, such as Cu–CuSO$_4$ and Ag–AgCl electrodes (e.g. for corrosion monitoring). In many laboratory studies, however, the electrodes are designed and manufactured in-house and the construction details are rarely reported (Leinov and Jackson, 2014; Vinogradov et al., 2010). These electrodes are typically designed only for use in a specific experimental apparatus.

The emergence of 3D printing technology has enabled the adaptation of sensors, featuring repeatability, precision, and mechanically useful parts, with applications across a range of research fields (Adamski et al., 2018; Ni et al., 2017). Kings College's FreeStation (KCL, 2021) is a repository for a variety of sensors and parts, and the technology has been used to develop small-scale reference electrodes for medical and other small-scale applications (Rohaizad, 2019; Schuett, 2021). Furthermore, printed junctions (ceramic disks) have been demonstrated (Sibug-Torres et al., 2020). In this paper, the design and construction of a simple, versatile, rechargeable, printed Ag-AgCl reference electrode is detailed. This electrode is suitable for laboratory-based SP measurements and is based on the design used by Vinogradov et al. (2010). The electrode consists of an Ag rod with a chloride coating, a chloride salt reference electrode, a porous ceramic membrane to allow charge exchange with the experimental electrolytes and an enclosure. The Cl$^-$ ions in the reference electrolyte are in equilibrium with the Ag rod such that (Jackson, 2015):

$$AgCl + e^- \rightleftharpoons Ag + Cl^- \tag{1}$$

The use of 'rechargeable' here means that the Ag rod and associated reference electrolyte can be replaced periodically. This is important because ion exchange with experimental electrolytes causes the composition of the reference electrolyte to change



over time, which then causes the ion exchange with the silver surface to deviate from the simple equilibrium expressed in
equation (1). This in turn causes the liquid junction potentials across the membrane, and the polarization at the silver surface, to change. When this occurs, the measured electrical potentials typically become unstable, with large and rapid fluctuations and drift. The silver rod must be then removed from the electrode, have its chloride coat stripped and replaced, and the reference electrolyte must be refreshed (e.g. Vinogradov et al., 2010). These requirements impose constraints on the electrode design. Therefore, as shown in Fig 1, a mechanism to release the electrode is also included and is one of the advantages of this
configuration. The concentration and composition of the reference electrolyte may also be modified, depending on the experiment to be conducted.

This paper consists of five sections. An overview of techniques and motivation is presented in Section 1. The design and construction choices are laid out in Section 2. The testing methods are described in Section 3. Results confirming the accuracy of the Ag-AgCl reference electrode for different types of SP measurements are given in Section 4, followed by a summary and
75 future design developments and suggestions in Section 5.

## 2 Design and Manufacture

This section details the Ag-AgCl standard electrode on which the new 3D printed electrodes are based and lays out the specific requirements and manufacture of two different designs of printed, non-polarising Ag-AgCl electrodes. The designs presented here are based on the Ag-AgCl electrode reported by Vinogradov et al., (2010); this example design is referred to as the
Vinogradov Electrode (VE) and the aim of this work is to develop electrodes, based on the Vinogradov design, that are easily manufactured using 3D printing and can be used for a variety of SP monitoring purposes. Our new Electrode A is a copy of the VE, adapted for 3D printing; our Electrode B is a low-pressure, printed version of the VE adapted for flush mounting. This section is supported by design drawings in Appendix A and .stl (printable 3D design) files in Appendix B and available to download alongside this paper.





## 2.1 The Vinogradov Electrode

The Vinogradov Electrode has a 6-component, double female body design (Figure 2), which has two different threads (1/4in NPT and 1/4in BSPP connectors on the rear and front ports of the electrode, respectively); one is tapered and one is straight to ensure the probe body is connected correctly to the experimental apparatus. The 'rod cap', which seals into the rear port, uses a 2-part gland seal with an O-ring to seal the Ag-rod in place. The front port allows for a wide variety of connections to experimental apparatus through variation of the rod/line cap (shown in Figure 1).

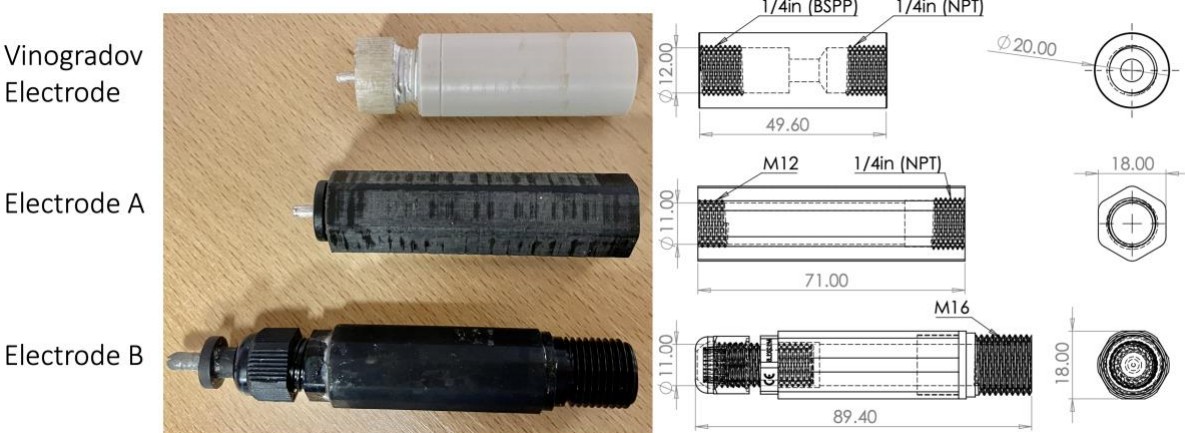

**Figure 2: A photo (left) and major dimensions of (in descending order) the Vinogradov Electrode, Electrode A and Electrode B, (right).**

The parts used in the Vinogradov design were either machined specifically for purpose (e.g. the electrode body and gland seals) or are produced commercially for a variety of applications (e.g. silver rod, O-rings and ceramic membrane). The electrode body is manufactured from PEEK, specified to handle high pressure (up to 0.5MPa) and temperature (up to 125°C) consistent with the experiments conducted using the electrodes by Vinogradov and co-workers (Al Mahrouqi et al., 2017; Collini et al., 2020; Vinogradov, Jan and Jackson, 2015) . The bespoke machining of the electrode body means they are expensive and require expert manufacture. Moreover, thread degradation and warping can require the application of considerable torque to disassemble the electrode, due to the round nature of the body, which can damage the body of the electrode. A flat-sided design, that could be gripped by a spanner, would reduce wear and tear on the electrode body. The Vinogradov design also incorporates several small O-rings which prove complex to install and may not be necessary for lower-pressure applications. It should also be noted that the Ag-rod is never fully submerged during the AgCl coating step (explained in Section 2.4), an insight which enables the development of a bonded Ag-rod adapter.





## 2.2 Design Considerations

3D printing offers great flexibility when designing non-polarising electrodes for specific applications. During this work, a direct replacement for the VE (Electrode A), and a modified version of the VE for flush mounting (Electrode B) were developed. The new electrodes can be summarized as:

- Electrode A – A 3D printed version of the VE (Female-Female body), which incorporates several design improvements and is able to withstand seal pressure differentials up to 0.5 MPa and fluid temperatures of 140°C.
- Electrode B - A flush mount version of the VE, optimised for usability but with a lower pressure rating of 1 kPa.

The combined design brief, in line with the VE, specifies that the electrodes must accommodate:

- a removable 3mm diameter Ag-rod;
- a 12mm diameter, 6mm thick, porous membrane that can be replaced for maintenance;
- easy cleaning and reuse, and
- improved useability compared to the VE.

These design constraints were adopted for both Electrodes A and B, as shown in Figure 2. The key differences between the newly developed electrodes and the VE are:

- *Tool Compatibility* – an 18mm hexagonal prismatic design was used and, to reduce fracture and damage, a 3mm fillet was applied to each of the hexagonal points.  Note that an octagonal design could have been used.
- *Reduction of parts* – the rod cap (Figure 1) was simplified into one part and fused to the Ag-rod using epoxy resin. This removes a step from the assembly stage.
- *Alternative porous membrane* - a sintered silica porous material (porosity rating 00) is used in the VE; here, a Polytetrafluoroethylene (PTFE) membrane was trialled and found to be interchangeable with the sintered glass. The PTFE membrane is cheaper and easier to machine to the correct dimensions, further reducing the cost of an electrode.

Electrode A follows the design of the VE and it is recommended to print in an ABS plastic (with a dissolvable support material – for printing purposes only) for high-pressure/temperature applications, and pressure test the electrodes prior to use to ensure there are no leaks. Electrode B is designed for low-pressure experiments and can be mounted flush against a tank wall. The thread-based Ag-rod gland was replaced by an IP68 cable gland to form a quick-release seal against the fused rod-cap electrode assembly (Figure 2). A concern at the time of manufacture was that flush tank tapping points may be prone to cracking, so it was decided to use a moulded Polypropylene (PP) tip for the electrodes to reduce the chance of electrode implanting or removal damaging the tank. The printer used in the current study was not able to print in PP (or Nylon); readers with access to more advanced 3D printers could consider removing this step and printing the entire body. A comparison of the three electrodes is given in Table 1.  Note that the cost of the printed electrodes presented here is of order x40 to x75 lower than the equivalent VE.



**Table 1: - Comparison of parameters and desired performance for Electrode A, Electrode B and the Vinogradov electrode. Note that ratings depend on the manufacturer and the authors accept no liability for electrodes not reaching these ratings. Prices shown are for body and electrode cape *VE price quoted in 2019, **Calculated in ABS/equiv.(inc. parts in Appendix A).**

| Parameter | Vinogradov Electrode | Electrode A | Electrode B |
|---|---|---|---|
| No, of electrode parts | 1 | 1 | 3 (inc. gland assembly) |
| Number of parts | 9 | 5 | 5 (inc. gland assembly) |
| Rod Cap fused to electrode | No | Yes | Yes |
| Spanner size [mm] | N/A | 18 | 18 |
| Cost of electrode parts | £111.21* | £1.46 | £2.76 |
| Max Electrode length [mm] | 54 | 65 | 65 |
| Electrode Diameter [mm] | 3 | 3 | 3 |
| Porous Disc Diameter [mm] | 12+/- 0.1 | 12+/- 0.1 | 12+/- 0.1 |
| Porous Disc Thickness [mm] | 6+/- 0.1 | 6+/- 0.1 | 6+/- 0.1 |
| Manufacture Technique | Die Cast/Injection Mould | FFF | SLA |
| Body material | PEEK | ABS | PP |
| Other materials | - | Epoxy Resin | Epoxy Resin |
| Connection Front | 1/4in NPT | 1/4in NPT | M16 |
| Connection Rear | 1/4in BSPP | M12 | - |
| Sealing Method Front | O-ring/PTFE tape | O-ring | O-ring |
| Sealing Method Internal | O-ring | O-ring/PTFE tape | PTFE tape |
| Sealing Method Rear | O-ring/PTFE tape | O-ring | Cable Gland |
| Sealing Method Electrode | O-ring | N/A | N/A |
| Max Temperature [C] | 125 | 125 * | 35 * |
| Max Gauge Pressure [MPa] | 0.5 | 0.5* | 0.001* |

## 2.3 Manufacture

There are a variety of 3D printing methods available to most consumers/researchers; during this work, different printers were available at different times. Two widely available commercial methods of printing were used: Fused Filament Fabrication (FFF) and Stereolithography (SLA). There are advantages to both, but the FFF, with its far wider usage, adaptability, and support, is recommended for future work.

Electrode A was manufactured by FFF on Ultimaker S3 and S5 models. This model of printer allows for dual printing, used here to print dissolvable support material. For the best possible performance characteristics, it is recommended to print in ABS at the highest quality (Engineering - 0.1mm layer height, with a 2 mm wall thickness and 80% infill). If possible, dissolvable Polyvinyl Adhesive (PVA) support material for the threads is advised, though the 'Breakaway Support' material from Ultimaker offers reasonable results. Printing with the alignment of the long axis of the electrode body to the vertical axis of the printer offers the greatest electrode detail; these options are available at the slicing (coding) step of print preparation. To finish the electrode, it is recommended to run a tap (threading tool) up and down the threads (in the case of Electrode A we





used a G1/4 and an M12). It is also recommended to apply a thin coat of print varnish or epoxy resin (with a low viscosity mix) to the interior, which aids in cleaning at later stages and reduces wear.

Electrode B has a hybrid construction, due to manufacturing constraints during this work. The tip and gland of the Electrode B are cable glands; the tip requires some work on a lathe to incorporate the membrane at the end of the tip. The body was printed using SLA, in this case an Anycubic photon S. When slicing a minimal layer height was preferable, as was printing along the vertical axis of the electrode. If threads are included in the design, it is recommended to run any thread cutting tools before the final curing process, as the material is less brittle at this stage. For the electrode B manufacture, once cured the body

was epoxied together with the tip and read gland. To complete the body of Electrode B, an M12 cable gland was bonded by epoxy resin to the rear end of the electrode body, and an M16 gland (as shown in Appendix A) was bonded to the front end.

### 2.4 Preparation of the electrodes

This section outlines the steps for the preparation of the electrodes prior to use, for the VE and our new Electrodes A and B. The steps are included as a reference and to give further background to the design considerations expressed above.

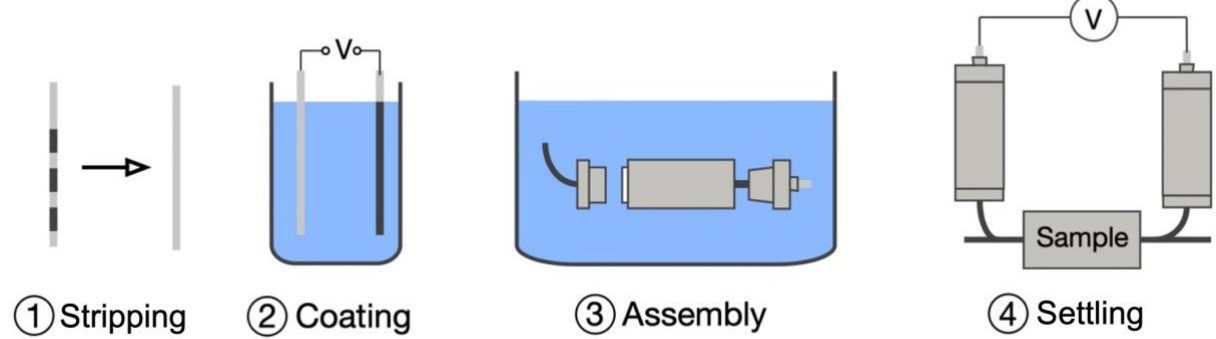


**Figure 3: The four stages of electrode preparation.**

The electrodes used here are prepared in a 4-stage process (Figure 3); the same preparation stages are followed when first assembling the electrodes after manufacture, and when they are cleaned and refreshed. First, the silver rod must be prepared: a low grit (120-240 grit) emery cloth is used to clean the silver surface; the surface must also be clear of grease and other

contaminants (Figure 3). It should be noted that fresh silver rods must also be cleaned and rubbed lightly with emory cloth. The rod is then placed in a NaCl bath, following the approach of Vinogradov et al., (2010) (a 1 M NaCl solution is used/recommended), along with a donor silver electrode; both rod and electrode are held in place using a crocodile clip and are not fully submerged. A current (of order 1 A at 12 V DC) is passed through the rod and donor electrode (normally for a few seconds) until a brown coating is evenly applied (step 2 of Figure 3). The (cleaned and washed) electrode body is then

assembled, and the porous membrane disc is inserted; prior to insertion, the porous membrane disc should be soaked in electrode electrolyte solution until saturated.



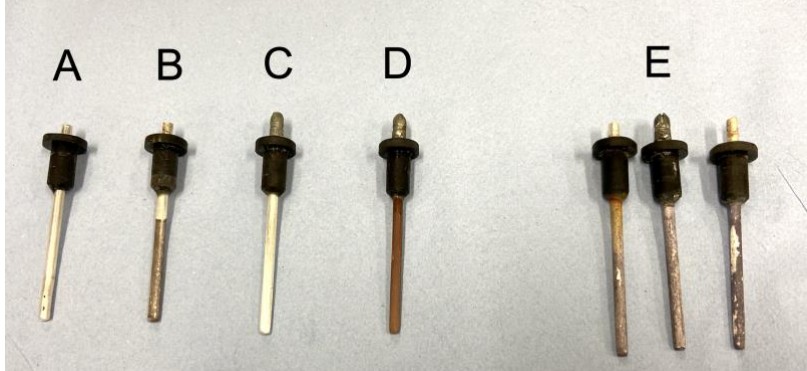

**Figure 4: A photo showing the stages of Ag-rod refresh – (A) a rod ready for coating, (B) an undercoated electrode, (C) an unclean/stripped Ag-rod where recoating has failed (D) A correctly coated rod (E) a collection of degraded Ag-rods after use.**


The electrode is assembled in step 3 in a bath of the electrode electrolyte solution to ensure no air bubbles are trapped inside the electrode. Finally, the electrode is 'settled' by monitoring the voltage(s) between 2 (or more electrodes) over a period of time; this can be across an experimental sample, or in a low ionic strength electrolyte bath (e.g. tap water). Once made up, it is important to ensure that there are no leaks and that the electrode membrane (the porous disc) remains wet. As shown in

Figure 4, care must be taken when preparing the Ag-rods; the rods shown under (E) show various evidence of degradation, including discoloration and flaking. Only rod (D) shows a correctly stripped (A) and prepared Ag-rod (D). Example (B) shows the dangers of not thoroughly stripping the Ag-rod, while example (C) demonstrates inadequate time/current when coating the Ag-rod.

## 3 Electrode Performance Testing

To demonstrate the accuracy and stability of the new 3D printed electrodes, they were used in two different laboratory experiments to measure the SP and their performance compared against previous data obtained using the Vinogradov electrode. In the first set of experiments, the electrodes were used to measure the streaming potential, which is the component of the SP that arises in response to a pressure gradient across an electrolyte-saturated porous medium (e.g. Jackson, 2015). In the second set of experiments, the electrodes were used to measure the exclusion-diffusion potential, which is the component of the SP

that arises in response to component concentration gradients across an electrolyte-saturated porous medium (e.g. Jackson, 2015). All testing was conducted using electrodes prepared with an Ag rod of length 65mm and 3mm diameter, coated in AgCl, as detailed above, and filled with 0.63 M NaCl electrolyte. The experiments were conducted to confirm that the new electrode design and materials used ddo not interfere with the electrical signals.



### 3.1 Streaming Potential Measurements

The experimental apparatus is comprised of a metallic core-holder with non-metallic end-caps, within which the cylindrical rock sample ('core') is held inside a rubber sleeve; this ensures there is no electrical contact between the sample and the metallic parts of the core-holder (Figure 5). The core-holder is engineered to apply a confining pressure which ensures that fluid is forced to flow through the sample rather than between the sample and core-holder. Flow lines pass through the endcaps and can be used to cause fluid to flow through the sample. Two SP electrodes are connected to the flowlines on either side of

the core-holder and out of the direct path of the flow, to measure the voltage difference across the sample. Each flowline is connected to a reservoir containing the electrolyte of interest and a mineral oil containing a red dye (Figure 5a). A pump is used to induce electrolyte flow through the core, using the mineral oil as a hydraulic fluid to force the electrolyte out of the inlet reservoir and through the sample to the outlet reservoir, creating a pressure drop across the sample.

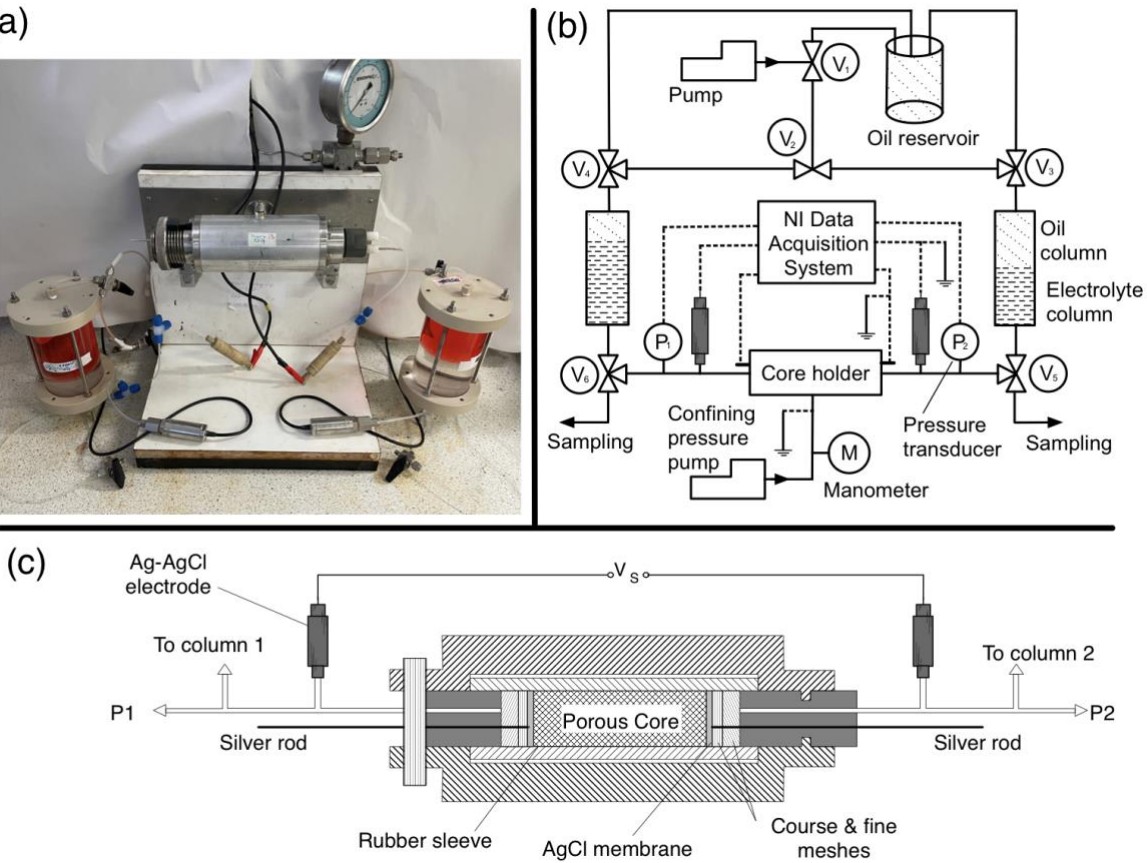

**Figure 5: A photo (a), hydraulic diagram (b) and core cross-section (c) of the experimental apparatus used for measuring the streaming potential coupling coefficient of a porous core sample. The electrolyte is pumped through a saturated porous core sample shown in (c), using the hydraulic system shown in (b), while pressures and self-potential voltage ($V_s$) and voltages from internal electrodes are recorded – here on a National Instruments high impedance differential voltage logger. The electrolyte can be pumped in both directions through the core, through the manipulation of the 6 valves (V1-V6). For further details please see Jaafar et al.,**

**2009.**





Stabilised pressure and voltage measurements are recorded for several different but constant flow rates, and with flow reversed to ensure the pressure and voltage responses are symmetric with respect to flow direction. Plotting the stabilised voltage difference against the stabilised pressure difference for each flow rate allows determination of the streaming potential coupling coefficient ($C$), given by the gradient of a linear regression through the pressure and voltage data. The zeta potential, which is

a measure of the electrical potential on the surfaces of the porous sample, can then be calculated using the Helmholtz-Smoluchowski equation (e.g. Collini et al., 2020; Li et al., 2016). More information on the experimental process can be found in Vinogradov et al., (2010).

### 3.2 Exclusion-Diffusion Potentials

Two apparatus are used in experiments to measure the exclusion-diffusion potential across porous samples (Leinov and

Jackson, 2014). The approach accounts for electrode effects, which dominate the exclusion-diffusion potential. The electrode potential arises because the electrodes are in contact with electrolytes with different composition.

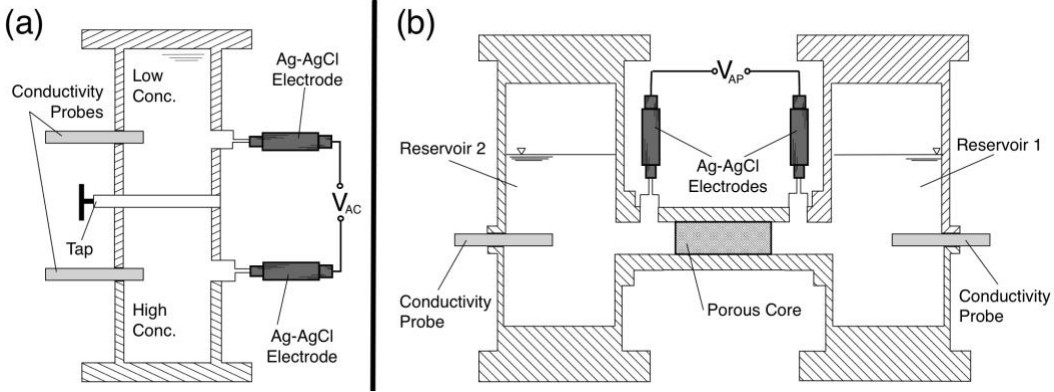

**Figure 6: Cross-sectional sketches of the apparatus used to measure the exclusion-diffusion potential across a saturated sample (a**

**'plug'): (a) shows the 'plug apparatus' for measuring the exclusion-diffusion potential across the sample; (b) shows the 'column apparatus' for measuring the diffusion potential across the two electrolytes of interest.**

The 'column' apparatus (Figure 6b) is used to measure the diffusion potential across the two electrolytes of interest, including the electrode potential. The diffusion potential is then calculated for the two electrolytes and subtracted from the measured potential to determine the electrode potential. In this apparatus, two reservoirs are connected in a vertical arrangement, with

the upper reservoir filled with the electrolyte of lower total concentration (and therefore lower density), and the lower reservoir filled with the electrolyte of higher total concentration (and therefore higher denser). The interface between the two electrolytes is therefore gravity stable. Two electrodes, one in each reservoir, are connected to the apparatus. When the tap is opened, an interface is established between the two electrolytes, allowing ions to pass from one electrolyte to the other by diffusion. An electrical potential difference is established across the interface, which is measured by the electrodes. This potential is termed



the 'apparent column' electrical potential $\Delta V_{AC}$ and is given by the sum of the (unknown) electrode potential and the (known) diffusion potential $\Delta V_D$ for the two electrolytes of interest.

$$\Delta V_{AC} = \Delta V_D + \Delta V_C \,. \tag{2}$$

The plug apparatus (Figure 6a) is used to measure the exclusion-diffusion potential across the saturated porous material of interest. In this apparatus, the two reservoirs are each filled with one of the two electrolytes of interest (the same electrolytes used in the column experiment) and connected by a sample of the porous material of interest. The sample is pre-saturated with

the lower concentration electrolyte, and tightly confined to ensure there is no transport of electrical charge around the outside of the sample. As before, an electrode is located in each reservoir. As soon as the reservoirs are connected, an interface is established allowing ions to pass from one electrolyte to the other by diffusion through the porous medium. An electrical potential difference is again established across the interface between the two electrolytes, which is measured by the electrodes. This is the apparent plug electric potential $\Delta V_{AP}$ and is the sum of the (unknown) exclusion-diffusion potential and the electrode

potential determined in the previous experiment.

$$\Delta V_{AP} = \Delta V_{ED} + \Delta V_C \tag{3}$$

The unknown exclusion-diffusion potential across the saturated porous sample can then be determined (Leinov and Jackson, 2014). A more detailed description of the experimental method and apparatus can be found in Macallister et al., 2019). Here we consider only the column experiments that are used to establish the electrode behaviour.

### 3.3 Results of the Electrode Tests

This section presents results from the streaming potential and exclusion-diffusion potential experiments. During the exclusion-diffusion potential experiments, it was possible to use Electrode A and B interchangeably with the VE. However, for the streaming potential experiments, Electrode A was used due to the high differential pressures involved.

### 3.3.1 Streaming potential experiments

Figure 7 shows results from the streaming potential experiments obtained using a porous sample comprising packed silica

glass beads of 1mm size and two different electrolytes used in laboratory experiments of saline intrusion (Etsias et al., 2021 and see section 4). The electrolytes are tap water and synthetic sea water (Table 1).

**Table 2: Materials used in the experiments reported here.**

| Material | Sample Porosity | Electrolyte | Salinity equivalent to NaCl [Mol/l] | Electrolyte conductivity [mS/cm] |
|---|---|---|---|---|
| Glass beads of 1mm dia | 0.219 | Tap Water | 0.004 | 0.42 |
| | | Synthetic seawater | 0.643 | 51.27 |

Panels (a) to (d) report examples of the 'raw' data from each experiment, showing the pressure drop (panels a and b) and voltage (panels c and d) across the sample as a function of time. Separate experiments were conducted for each electrode type,

giving rise to the data reported in blue (VE) and black (Electrode A).





The pressure data show good reproducibility across the two experiments, with each increase in pressure drop corresponding to an increase in the flow rate across the sample induced by the pumps. After each flow rate change, the pressure reaches a new stable value. The voltage data recorded by the different electrodes also show similar behaviour, with the voltage decreasing (becoming more negative) each time the pressure drop increases. After each flow rate change, the voltage reaches a new stable

value. Plots (e) and (f) record the stabilised voltage plotted against stabilised pressure for each flow rate, for the experiments with the different electrodes and the two different electrolytes. As discussed previously, a linear regression through these data yields a key property, the streaming potential coupling coefficient (*C*).

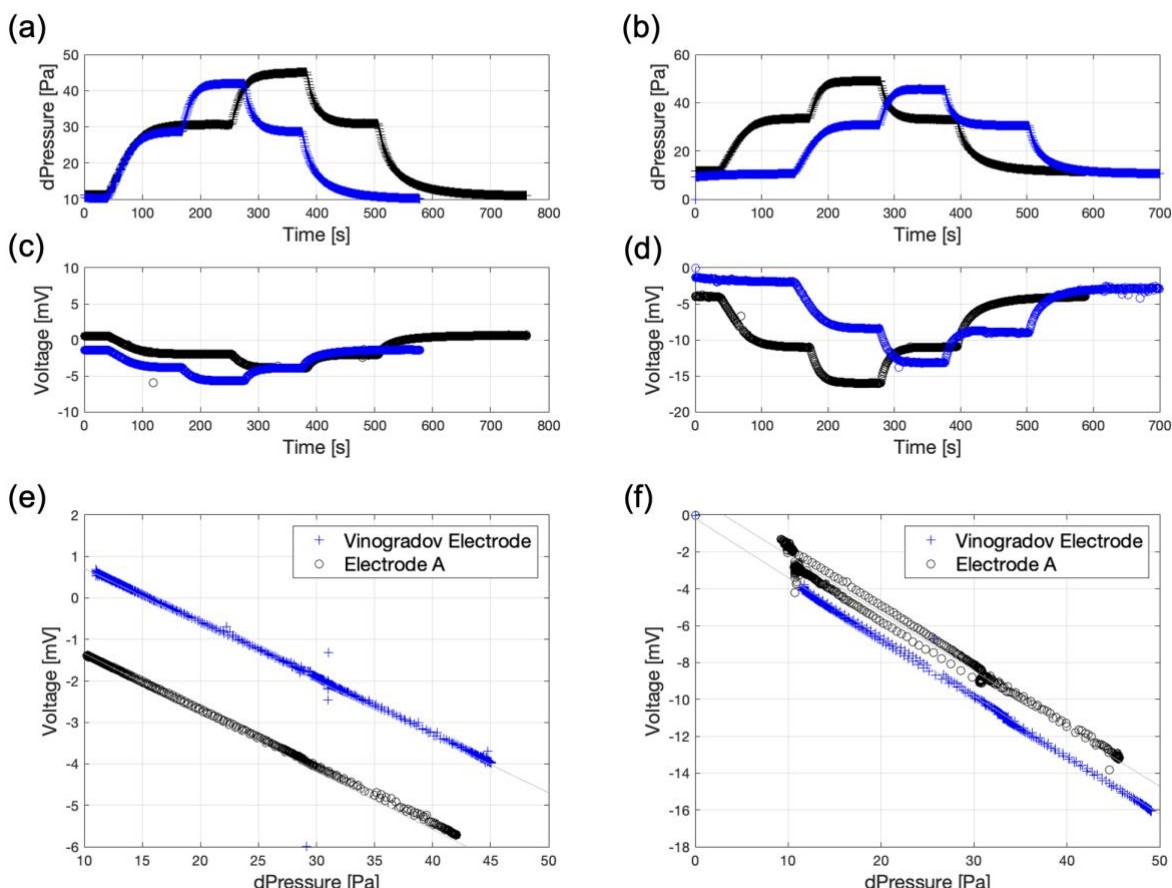

**Figure 7: Streaming potential results for the glass beads obtained using synthetic seawater (left) and tapwater (righ) comparing results from the VE and the developed Electrode A. Plots (a) and (b) show pressure change over time; plots (c) and (d) show voltage change over time; and plots (e) and (f) show stabilized voltage against stabilized pressure drop.**

The coupling coefficient for the synthetic seawater electrolyte is measured to be 154 mV/kPa using the VEs, and 142 mV/kPa using our new Electrodes A, a difference of 7.8%. The coupling coefficient for the tap water is measured to be 323 mV/kPa

using the VE, and 320 mV/kPa using our new Electrodes A, a difference of just 1%.



### 3.3.2 Exclusion-diffusion potential experiment

Figures 8 and 9 show results from the exclusion-diffusion potential measurements using the same two electrolytes: synthetic seawater and tapwater. We report here only the column experiment used to determine the electrode potential; the recorded electrical potential as a function of time is shown in Figure 8.

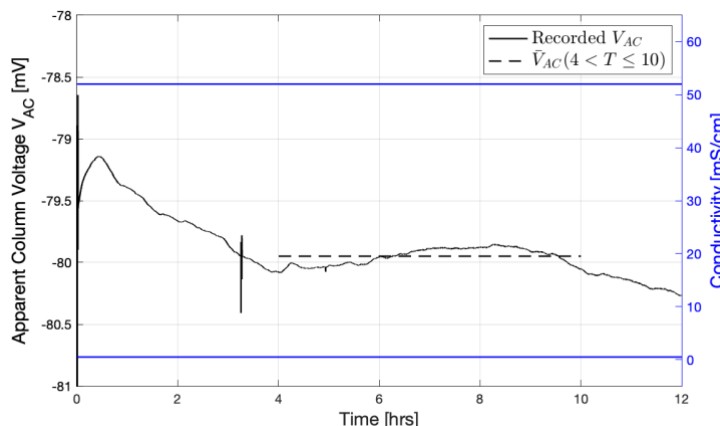

**Figure 8 : Results from a column experiment using the tapwater and synthetic seawater electrolytes described in table 2, two electrode Bs were assembled using synthetic seawater. The figure shows the recorded apparent column voltage recorded against time, also plotted is a mean voltage from t = 4 to 10 hours as a dashed line; the right-hand axis shows the conductivities at the top and bottom of the column (in blue).**

There is an initial period up to 4 hours during which the measured voltage varies as the electrodes equilibrate with the electrolytes, followed by a period of stable voltage up to 10 hours, after which the local electrolyte concentration around the

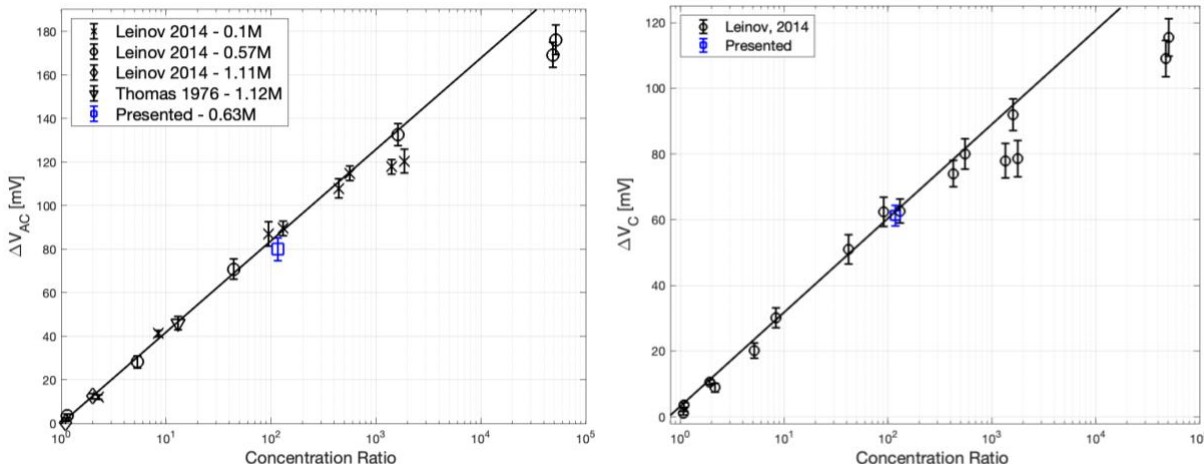

**Figure 9: Summary of the results from the column experiment. (a) Stabilized voltage as a function of concentration difference along the column, for a fixed temperature of 298 K (20°C). Also shown is a single data point from a similar experiment by Thomas [1976], and the data from Leinov and Jackson [2014] which was used to create the linear regression. (b) Electrode response calculated from (a) and compared to the data from Leinov and Jackson [2014] which was used to create the linear regression. The gradient of the linear regression yields the electrode concentration sensitivity.**





electrodes begins to change. The chosen stabilized voltage is shown as the dashed line, to give the apparent column voltage of 79.95mV. The diffusion potential of the two electrolytes used was calculated using the approach outlined in MacAllister et al. (2019) and found to be 18.7 mV; the electrolyte concentration ratio was 117.. Figure 9a shows the stabilized voltage

plotted against the electrolyte concentration ratio, and previously published experiments reported in Leinov and Jackson (2014). Figure 9b shows the corresponding electrode potential, obtained after subtracting the diffusion potential. Error bars denote the uncertainty in identifying the stable electrical potential. We observe an excellent match between the new measured data and the published data. The published data report an electrode sensitivity (the gradient of the linear regression M to the data plotted in Fig 9b) of 30.3 mV/decade, compared to our measured electrode sensitivity of 29.6mV/decade, a 2.2% difference.

## 285 4 Application to saline intrusion monitoring in a laboratory experiment

The previous section demonstrated that the printed electrodes presented here provide comparable performance to the reference Vinogradov Electrode. To demonstrate the flexibility of the rapid printing method when constructing electrodes, this section briefly details the use of Electrode B to monitor SP in a large tank experiment designed to replicate the 'Henry problem' (Henry, 1964) Saline water ('synthetic seawater') invades a porous medium (silica beads) saturated with fresh (tap) water (see

Table 2 and (Etsias et al., 2021; Robinson et al., 2016)), creating a wedge of saline water along the base of the porous medium. The experimental tank measures 1 m x 0.6 m x 0.014 m and was designed to be instrumented with 12 electrodes Bs mounted flush on the 25 mm thick acrylic rear wall of the tank (Figure 10).

The electrodes were recharged with the 'synthetic seawater' electrolyte and mounted into the rear of the tank as shown in Figure 10. The SP electrodes were referenced to the upper-leftmost electrode (F1); all electrodes were assigned alphanumeric

codes with a letter detailing their row and the number of their column in the array. From the right-hand-side of the tank (nearest to A column of electrodes), the synthetic seawater was intruded into the tank in 3 stages corresponding to 20, 16 and 14 mm

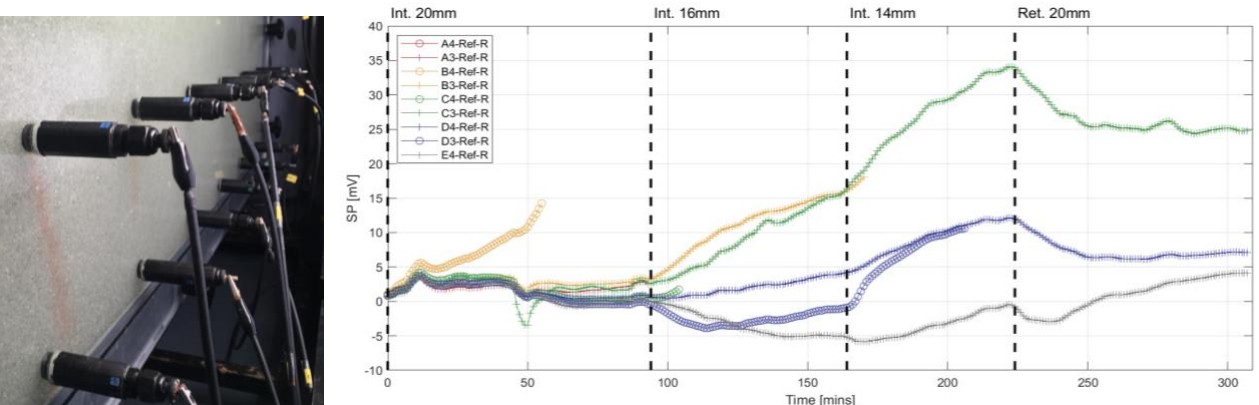

**Figure 10: A photo (left) and recorded SP signals (right) from flush mounted Electrode Bs made up with 0.63M 'syntenic seawater', in a homogenous synthetic aquifer made of 1.09mm silica beads, undergoing saline intrusion when 'synthetic seawater' displaces freshwater and then retreats.**

head differences, and then retreated away in the fourth phase, returning to a 20 mm head difference (Etsias et al., 2021). The



saline-freshwater interface was allowed to reach equilibrium in each phase. These phases are marked in Figure 10. Note that when an electrode was overcome by the saline electrolyte, the recordings are omitted from Figure 10.

The experimental results and their interpretation using numerical simulations are beyond the scope of this paper; what is of interest here is the stability and smoothness of the recorded SP signals. As shown in Figure 10 (especially in the first 50 minutes of the experiment), the SP signals track each other, demonstrating a high degree of stability and replicability of the measurements. The low level of noise and the reversal of signals during the retreat phase (after 224mins) further demonstrates that the electrodes are stable and sensitive to the self-potential signals generated in the experiment.

**5 Conclusion**

We have reported the development of two simple, robust, and stable printed non-polarizing electrodes for self-potential monitoring that can be modified in design for high and low-pressure experiments. The electrode refreshment process described above is critical to the stability of the electrodes; an aggressive abrasive stripping of the silver electrode is strongly recommended. The electrodes are compatible with many experimental fittings, including flush mounting and are

considerably cheaper to manufacture than the reference electrode against which their performance was compared. When designing probes, fully understanding the use cases is key, including pressure requirements, length of experiment, etc. This allows users to reduce over-design. A quick electrode removal system, using a cable gland, was trialed and found to be reliable. None of the materials used in the various manufacturing techniques used here were found to interact or affect the stability of the results. When manufacturing electrodes, printing all in a vertical orientation is strongly recommended; and if

including threads to a design, it is highly advisable to have the appropriate (threading) tap. Full details, designs, and further materials can be found in the appendices (including 3D print .stl files). Further parametric studies, removal of the membrane (through integrated printing), and other refinements will be the focus of future work, as well as the experiments that these electrodes were designed to monitor.

**Author Contribution:** Rowan, designed and built the electrodes. Rowan and Karantoni tested the electrodes, reporting to
Jackson and Butler. Jackson and Butler interpreted the results, all co-others contributed to the preparation of the manuscript.

**Conflicts of Interest:** The authors declare that they have no conflict of interest.

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
