# Peer review of "3D-printed Ag-AgCl Electrodes for Laboratory Measurements of Self-Potential"

_EGUsphere, 2023_

## Author Response (AR1)

14th August 2023 To: Dr. Lev Eppelbaum Department of Geophysics Tel Aviv University

Dear Dr Eppelbaum,

Thank you very much for your thoughtful feedback on our manuscript. We have made the minor changes requested by the referees, I enclose a summary of these changes and the rebuttal posted on the 9th of August.

We greatly appreciate your time and effort to help us improve our work.

Yours Sincerely,

Thomas Rowan (on behalf of the authors)

**Rebuttal 9th of August**

We are disappointed that the reviewers recommend that the paper should be rejected and do not believe that their comments provide grounds for this decision. The reviewers find no flaws or errors in our work and seem to miss the point of the paper.

RC1 suggests 'the purpose, importance and innovation of this research are not elaborated'. We disagree. As we point out in the paper, the purpose is to demonstrate a practical method for the manufacture of non-polarizing electrodes that is quicker, cheaper and more versatile than traditional approaches. The importance is that these electrodes are traditionally expensive and slow to construct. Saving on experimental costs and time is important to maximize the value delivered from research income. The innovation comes in the use of 3D printing.

RC1 further states '3D printed method is seemed as an innovation, but not be illustrated in the paper. It just like the authors did one job and wrote it down. The authors have finished the job, but did not find the scientific meaning.' Again, we disagree. We demonstrate the electrodes give high quality SP data and, importantly, provide detailed design sketches, 3D printing files, and practical hints and tips for construction. Interested readers can use these to create their own electrodes and use these in experiments to find – as we have done in other work – the scientific meaning from high quality SP data.

RC1 concludes: 'The structure of this paper is also not that appropriate that is more suitable for a patent or trial report. And some words are confused like "Designs are included for both male and female connections to laboratory equipment".' We have taken great care to structure the paper in line with other papers in GI, aiming to serve both as a scientific journal paper and a valuable reference tool for the community that will continue to develop these sensors. The sentence quoted to support the claim that 'some words are confused' makes use of commonly used terms for connectors.

RC2 claims the 'only innovation concerns the 3D printing of the (Ag-AgCl electrodes)'. Indeed, that is the innovation and we believe it will be of interest to the GI journal readership. They state 'Most of the paper consists of a description of the well-known experiments, which must verify the good performance of the new electrodes.' Some of the paper is indeed devoted to describing these experiments; as we explain in the text, we need to demonstrate that the new electrodes deliver high quality data, comparable to those constructed using traditional methods. They state 'It should also be noted that the use of 3D printing is supposed to considerably lighten the manufacturing process, however, we must always intervene on the printed body with tapering tools or lathe.' We agree, but these final finishing steps to the 3D printed bodies is still considerably less time consuming than traditional manufacture and, as we show, the cost of producing our printed electrodes is much lower. The reviewer concludes 'I think that all of the new information included in this paper are not adapted for a scientific publication in the GI but may deserve a publication of a note in a more technical journal.' We respectfully disagree with this view, and believe the content is well suited for publication in GI.

The reviewer suggests some minor corrections which have been implemented.

We have revised the paper to address these minor corrections, and also emphasize that we report design drawings, practical advice for electrode printing and assembly, and include printable 3D design files to facilitate wide uptake.

| Location | Comment                                    | Correction                                          |
|----------|--------------------------------------------|-----------------------------------------------------|
| L15      | "Nyquist and Corry, 2002" is not given in  | Thankyou this has been added.                       |
|          | the references.                            |                                                     |
| Figure 1 | Add on the fig.1 description "moulded      | This has been clarified in the text (the design in  |
|          | Polypropylene (PP) tip" designation.       | Figure 1 is not commonly made in PP).               |
|          |                                            |                                                     |
| L141     | Add the manufacturer info of the 3D        | Thankyou, these have been added.                    |
|          | printers in the refs.                      |                                                     |
| L178     | "over a time" how long?                    | Thankyou we have included our standard 24hrs        |
|          |                                            | in the text (though it can depend on the makeup     |
|          |                                            | of the electrode).                                  |
| L180     | "show various evidence of degradation" due | Thankyou, this has been clarified in the text, to   |
|          | to?                                        | explain the degradation cause.                      |
| L193     | "ddo" should be "do".                      | Corrected.                                          |
| L269-270 | why the differences are so remarkable      | This has been clarified in the text, it is expected |
|          | between synthetic seawater and tap water   | and in line with other studies.                     |
|          | electrolytes experiences?                  |                                                     |
| L273     | and Figure 8 legend "tapwater" should be   | Corrected.                                          |
|          | "tap water".                               |                                                     |
| L279     | "" should be "."                           | Corrected.                                          |

**Minor Corrections from RC2**